# Transforming the Global Plastics Economy: The Role of Economic Policies in the Global Governance of Plastic Pollution

**Diana Barrowclough [1,\*] and Carolyn Deere Birkbeck [2]**

[1] United Nations Conference on Trade and Development, Palais des Nations, 1211 Geneva, Switzerland
[2] Global Governance Centre, The Graduate Institute Geneva, Chemin Eugène-Rigot 2, 1202 Geneva, Switzerland; Carolyn.deere@graduateinstitute.ch
\* Correspondence: Diana.Barrowclough@unctad.org

**Abstract:** International policy discussions on plastic pollution are entering a new phase, with more than 100 governments calling for the launch of negotiations for a new global plastics agreement in 2022. This article aims to contribute to efforts to identify effective international policy levers to address plastic pollution. It takes stock of the evolution of views and perceptions on this complex and multi-faceted topic—from concerns about marine pollution and waste management towards new strategic directions that involve the entire plastics life-cycle and include climate and health impacts associated with the proliferation of plastics. It also traces the progressive development of responses—from voluntary approaches involving multiple stakeholders to national and international approaches focused on regulation. The paper is informed by desk research, a literature review and participation by the authors in informal and formal global governance processes on plastic pollution, the environment and development in the United Nations and World Trade Organization between 2019 and 2021. It also draws on empirical findings from a novel and original database on the life-cycle of plastic trade created by the authors. The paper argues that the important focus on downstream dimensions of plastic pollution—and strategies to address them—needs to be complemented by a broad life-cycle and "upstream" perspective that addresses plastic pollution at its source. It highlights the political economy tensions and inconsistencies at hand, observing that while some countries are taking concerted efforts to reduce pollution (including through bans on certain kinds of plastic and plastic products); to promote more circular plastic economies; and to reduce the carbon footprint of plastics (as part of a wider effort to decarbonize their economies), trade and investment in the plastic industry continues to rise. The paper argues that to reduce plastic pollution, emerging global governance efforts must integrate international environmental law and cooperation with a complementary and enabling global framework that addresses the economic, financial, industrial and trade policies needed to drive the necessary transformation of the plastics sector.

**Keywords:** plastic pollution; plastic policies; plastic waste; circular economy; development; environment; trade; global governance



## 1. Introduction and Approach

Despite growing alarm about plastic pollution, the production and use of plastics is forecast to continue to expand over coming decades. Efforts on the part of governments, civil society and business to reduce plastic pollution are encouraging signs of awareness and an appetite for engagement but are, nonetheless, failing to stem the tide. This paper takes up two inter-related questions: first, how are views about the scope and causes of plastic pollution evolving, and second, what are the implications for how governments and stakeholders address these through international policymaking?

At present, there is a growing range of proposals for how international environmental law and policy could be strengthened to tackle plastic pollution, and especially marine plastic pollution (Haward 2018; Simon and Schulte 2017; Raubenheimer and McIlgorm 2018;

Tiller and Nyman 2018; CIEL 2020; Simon et al. 2021; WWF et al. 2020). During 2021, more than 100 governments indicated their support for the launch of negotiations on a new global agreement on plastic pollution in 2022. Some governments and stakeholders emphasise plastic pollution in the marine environment as the core priority for a potential new agreement, highlighting the importance of international cooperation on the critical task of improved "downstream" waste management and clean-up efforts. However, over 80 governments have signed a Ministerial Statement calling for an agreement that tackles plastic pollution across the life-cycle and incorporates a focus on tackling "upstream" drivers of plastic pollution.[1] While interest in the politics of plastics regulation and governance, there has been little systematic political or economic assessment of the range of global policy, regulation and governance options to tackle the upstream causes of expanding plastic production and pollution, as well as the wider development challenges associated with plastic pollution in developing countries and their efforts to reduce it.

This article aims to fill an important gap in the literature by highlighting the importance of coupling international environmental cooperation with enabling international economic policy frameworks to tackle the forces driving plastic pollution and to transform our economies away from the grip of plastics. In so doing, it aims to contributes an integrated analysis of evolving global discussions of challenges and responses to plastic pollution, offering an international political economy perspective.

The article begins by sketching the evolving framing and layered global discourse on plastic pollution and plastics policy, starting with early and persistent concerns about marine plastic pollution, moving to alarm about microplastics and toxic chemical pollution associated with plastics, and widening to broader concerns about plastic pollution across the entire life-cycle of plastics production, consumption and trade, including impacts on the climate). It highlights the growing attention to the potential for circular economy approaches to address aspects of plastic pollution and the need for "upstream" approaches to regulating the plastics industry as vital to tackling the downstream challenges of plastic pollution (Nielsen et al. 2020). In so doing, the article argues that attention to the political economy of the plastics industry will be central to advancing efforts to reduce plastic pollution. Recent findings on the breadth, volume and value of global trade across the lifecycle of plastic are used to illustrate the tremendous economic stakes at hand for many countries and companies. Further, the article emphasises the neglected issue of the relevance of the plastics industry to development in many developing countries, arguing that understanding the political economy of this growing industry in developing countries will be vital to its transformation and a just transition.

The paper then explores how international policy discussions and frameworks could better address upstream dimensions of the global plastics economy, emphasising the need to integrate attention to the ways that international economic policy frameworks, such as on finance, trade and industrial policies, will be needed to support and complement international environmental frameworks to reduce plastic pollution. Finally, the article contributes learnings for the future gathered from the authors' participation in evolving policymaking processes in the United Nations and World Trade Organisation and highlights gaps in knowledge for future research.

The analysis in this article draws on a review of the publicly available literature and official sources from international organisations as well as participation in a series of formal and informal processes on plastic pollution, trade and the environment taking place at the United Nations and World Trade Organization (WTO) during the period 2019–2021. These included workshops and conferences, informal dialogues and discussions, including the UNCTAD Quadrennial conference in 2021; the series of meetings held in relation to the Informal Dialogue on Plastic Pollution and Environmentally Sustainable Plastics Trade at the WTO and others associated with the United Nations Environment Programme (UNEP); the Conference of the Parties of the Basel Convention on Control of Transboundary Movements of Hazardous Wastes; as well as consultations with a range of government officials, policymakers, business and civil society groups engaged in debate on plastic

pollution. Working with chemical experts and using official United Nations Comtrade statistics, we created a database of the scale and volume of global trade across the life-cycle of plastics (see Barrowclough et al. 2020). Through these efforts the authors aim to contribute a useful and timely global policymaking perspective to the literature on the governance of plastic pollution.

## 2. Evolving Definitions of the "Problem"

### 2.1. An Overview of the Framing of the Plastics Crisis

Today's heightened public consciousness about plastic pollution has emerged gradually over several decades, accompanied by important evolutions in perception and understanding of the scope and causes of the problem. In terms of scope, the framing of the environmental dimensions of the plastics crisis can be organised into three broad layers that are also reflected in international policy discussions (Section 3). These categories are not necessarily consecutive and there are important feedback loops between them.

In the first category, the focus is squarely on marine plastic pollution. Findings from conservation groups, media and scientists show that in addition to larger pieces of plastic debris and litter, even the smallest microplastics are extremely damaging to marine biodiversity, ecosystems, wildlife and fisheries (UNEP 2021; GESAMP 2015; GRID-Arendal 2020; Jambeck et al. 2015). The scientific literature identifies numerous sources of plastic pollution in the marine environment and multiple strategies to prevent leakage into oceans and waterways (Boucher and Friot 2017; Boucher et al. 2020; Eriksen et al. 2014; Halpern et al. 2008; Quantis and EA 2020; The Pew Charitable Trusts and SYSTEMIQ 2020; UNEP 2016, 2020; Vince and Stoett 2018; WWF 2019).

Increasingly, studies of the growing tide of marine plastic pollution also have emphasized its economic implications and dimensions. The economic costs of marine plastic pollution, especially for developing countries, include the impacts on the environment, waste management systems, and local tourism, agriculture and fisheries (Barrowclough and Vivas-Eugui 2020; Schröder 2017, 2018; GAIA and Greenpeace East Asia 2019; Williams et al. 2019; WWF 2021). Many small island developing states, for instance, find themselves engulfed in a tide of plastic they cannot manage and which is damaging their fresh water supplies, overburdening already inadequate waste management systems, and damaging fisheries and tourism sectors that rely on healthy marine environments. Alongside, there is recognition that the growing tide of marine plastic pollution—and the fact that only a very small proportion of plastic waste generated annually is recycled—also reflects a significant loss of material value to the economy. The cost of failing to recycle or recapture the value of plastic packaging material used only once, for instance, has been estimated at some USD 80–120 bn in lost value to the global economy annually (WEF et al. 2016).

The second category broadens the frame to encompass the linked environment, health and human rights impacts of pollution in both marine and terrestrial environments, as well as in the air, across the life-cycle of plastics (Pellow 2007; Steensgaard et al. 2017; OHCHR 2021). This wider framing includes a focus on communities exposed to the environmental impacts of petrochemical and plastics production and manufacturing; toxic and hazardous chemicals present in plastic products, packaging and waste; and the health impacts of toxic chemical additives and microplastics that make their way into the food chain (Azoulay et al. 2019; Gallo et al. 2018). This frame emphasizes the many environmental and health risks associated with growing volumes of plastic waste (including the impacts of landfills, incineration and recycling that are not environmentally sound on neighboring communities and workers engaged in those sectors), as well as with the proliferation and expanding use of plastics for a broadening array of applications, including in the agricultural sector and production of food (FAO 2021).

The third broad category draws attention to the contribution of plastics to the climate crisis (Stoett and Vince 2019; Azoulay et al. 2019). More than 98% of primary plastics are based on fossil fuel feedstocks (widely known as 'virgin' plastics), meaning the plastics sector is an important contributor to carbon emissions (Material Economics 2018a). Each

stage of the life-cycle of plastic has impacts on the climate that will need to be addressed as part of wider international efforts to decarbonize the global economy, including: extraction and transport of fossil fuels for plastic production; refining and manufacturing; and waste management, reuse and recovery (Azoulay et al. 2019; CIEL 2017a, 2017b). If current trends in plastic demand continue as expected, $CO_2$ emissions from the plastics sector are forecast to rise by as much as 90% by 2050 (Wood Mackenzie 2021). By the same year, the plastic sector is projected to account for 20% of total oil consumption and 15% of the global annual carbon budget by 2050 (CIEL 2017a; CIEL et al. 2019; WWF 2019). While some plastic applications can support climate outcomes (e.g., light-weight plastic packaging can reduce the carbon footprint of international transportation; plastic car-parts can reduce the overall weight of cars and their associated fuel consumption; plastic insulation can help energy conservation), the plastics industry faces growing pressures for decarbonization, especially given its direct dependence on fossil fuel inputs and carbon footprint (Bauer and Nielsen 2021; Wood Mackenzie 2021; CIEL et al. 2019).

### 2.2. Perspectives on the Drivers of Plastic Pollution

These three broad frames on the scope or nature of "plastics problem" are accompanied by different views on the main drivers or causes of plastic pollution. Teasing these out is important because it impacts on subsequent perspectives on the appropriate responses.

### 2.2.1. A Global Waste Management Problem

A core explanation for plastic pollution, especially in the marine environment, is that countries lack capacity for environmentally sound waste management. At present, countries use a variety of strategies to collect, sort, recycle, incinerate, bury or reuse plastic waste, but these fall far short of the ever-growing volume of plastic waste. Many countries, especially developing countries, lack the infrastructure, fiscal resources and affordable access to technologies needed for environmentally sound waste management. In many developed countries too, capacity for environmentally sound recycling falls far short of growing needs, and most waste is either incinerated, buried or exported. Indeed, as developed countries have struggled to manage the rising volumes of plastic waste produced and used within their borders, a lucrative but problematic industry of plastic waste exports to developing countries has emerged, worth some USD 32.6 billion in 2019 (Marketsandmarkets 2019).

A subset of the waste mismanagement story focuses on developing countries, most of which lack the regulatory frameworks, institutional capacity and business infrastructure for environmentally sound management of plastic waste, and are among the top sources of plastic leakage into the oceans. Importantly, in addition to importing plastic products which become waste they cannot manage, some developing countries also produce plastic domestically; and some of the countries most associated with leakage into the oceans also import plastic waste from other countries.

This global political economy of the plastic waste trade only became prominent in the public eye when some countries started refusing plastic waste imports. Until 2018, China had imported over two-thirds of global plastic waste trade. After China adopted its "National Sword' policy, which restricted imports of plastic waste (Brook et al. 2018), less regulated countries such as Malaysia, Vietnam and Thailand as well as Indonesia, South Korea, Taiwan, India and Turkey took it in (GRID-Arendal 2019). Some of these countries also began to impose import bans and other measures, including sending unwanted plastic waste back to its country of origin (Ananthalakshmi and Chow 2019; Franklin-Wallis 2019).

From the 'waste mismanagement' perspective, central solutions to the plastics crisis are to expand investment in waste collection, management and clean up, including through investment in new environmentally sound recycling and incineration technologies, to support profitable international markets for recycling, and also to innovate and act to reduce the generation of plastic waste (European Commission 2018a). Alongside, governments have sought to strengthen the international framework for regulating global trade in plastic waste (see Section 3).

### 2.2.2. A Product Design Problem

Another argument is that a key driver of plastic pollution is the way products are designed and used, with single-use plastic products and packaging targeted as especially problematic. Among the key challenges identified are that products are not designed in ways that make them easily sorted or cost-effectively recycled (e.g., products that mix different types of plastic or that mix plastics with other materials such as cardboard and aluminium) and that the chemical composition of many plastics can make them not only toxic for human health and the environment, but also difficult or even dangerous to compost, biodegrade or recycle. Improved product design is a central component of campaigns for a more circular plastic economy, which aim to move businesses from a one-direction (take-make-waste) business model to a closed loop "circular" (take-make-take-make) business model that combine efforts to reduce plastic use with initiatives to reuse and recycle plastics (see for instance Ocean Conservancy and McKinsey 2017; WEF et al. 2016; EMF 2019b; OECD 2019a, 2019b; WRAP 2019).

From this 'product design' perspective, a central solution is to design plastics and derivative products in ways that reduce their environmental impact. From major producers of primary plastics to brands that use a range of single-use plastics, a diverse range of companies are partnering with or buying firms working to replace or mix conventional plastics with alternative plastics (such as recyclable, biodegradable, or compostable plastics); develop new product designs that limit the overall volume of plastic; reduce the range of materials used in ways that can facilitate downstream recycling; or end the use of specific plastics and toxic additives known to be especially harmful to the environment (Leslie et al. 2016; Royte 2019; WRAP 2019). However, not all of these pathways provide clear cut solutions, and the array of sustainability claims of many "eco-friendly" products are difficult to verify (Rucevska and Villarrubia-Gómez 2020). A 2015 UN Environment report concluded, for instance, that "the adoption of plastic products labelled as 'biodegradable' will not bring about a significant decrease either in the quantity of plastic entering the ocean or the risk of physical and chemical impacts on the marine environment . . . " (UNEP 2015). Similarly, there are concerns that the growing use of terms such as of "bioplastics" to refer to plastics made from renewable non-fossil fuel feedstocks can mislead consumers about their green credentials (Alaerts et al. 2018; Bauer 2018; Krieger 2019; Marshall 2007).

### 2.2.3. A Structural Problem

A third argument is that the plastic pollution crisis is a structural problem rooted in subsidised fossil fuels, expanding plastics production, and unsustainable consumption trends.

Underpinning this viewpoint is the fact that the growing global production and use of plastic far exceeds global capacity to manage the enormous volume of plastic waste generated (The Pew Charitable Trusts and SYSTEMIQ 2020). Alongside proliferating efforts to bolster international cooperation on plastic pollution, public and private investment in plastic production capacity is growing (American Chemistry Council 2018; The Guardian 2017; Barrowclough and Finkill 2021). Publicly-financed subsidies—from national, state and local governments, development banks and export credit agencies—have been identified as key factors keeping the price of fossil fuels relatively low and enabling the proliferation of cheap plastics on the global market (Skovgaard 2021; Steenblik 2021; Deere Birkbeck and Sugathan 2021). Moreover, as oil, gas and coal companies face regulatory and market pressures to shift the energy sector away from fossil fuel dependence, many are looking to petrochemicals and plastics as growth sectors (IEA 2018; Tullo 2019; Bauer and Nielsen 2021).

From this 'structural problem' perspective, key drivers of plastic pollution include the market and lobbying power of major multinational companies and state-owned enterprises active in the fossil fuels and petrochemical sectors. Following from this analysis, solutions to the plastic pollution will require attention to the international political econ-

omy of crisis (Clapp and Helleiner 2012; Dauvergne 2018a) and the pursuit of 'system change' approaches.

In 2020, research by The Pew Charitable Trusts and SYSTEMIQ concluded, for instance, that only a system change scenario could help reduce plastic leakage in the oceans to lower than current levels by 2040; noting that even that under this scenario some five million tonnes would still leak into the ocean each year. Their proposed system change approach requires ambitious and simultaneous implementation of multiple strategies focused on reducing and substituting plastics, increased recycling, and more environmentally sound disposal of plastics. Notably, they call for deployment of all available policy levers and technologies both downstream and upstream (The Pew Charitable Trusts and SYSTEMIQ 2020).

System change is not only central to reducing marine plastic pollution, but also the wider challenges associated with plastic pollution and its links to other environmental challenges: "The challenges of climate, plastic pollution and chemical toxicity—which at first might each seem like distinct problems—are actually interrelated and require a systems approach to resolve" (Mulvihill et al. 2021).

Further, the system change approach is grounded in an understanding that solving plastic pollution demands attention to reducing plastic production and consumption. Although many important and useful applications of plastic exist, including applications that can provide environmental benefits, a significant share of contemporary uses of plastics is not only environmentally harmful, but also excessive and unnecessary.

In terms of solutions, the structural problem argument includes calls for a significant share of plastic consumption to be eliminated, significantly reduced, replaced by substitute non-plastic products or omitted through the reintroduction or development of alternative delivery and business models. From this viewpoint, addressing the scale and multiplying dimensions of the plastic pollution crisis will demand greater engagement with the complex politics of reducing plastic production, ending fossil fuel subsidies, decarbonizing plastics, banning a range of toxic products, and shifting consumer behaviour away from unsustainable consumption of plastics—that is, on the politics of transforming the global plastics economy (Barrowclough and Deere Birkbeck 2020).

*2.3. Mapping Today's Policy Landscape—Examples of Evolving Responses and Approaches by Governments, Industry and Civil Society*

Just as the framing of the plastic pollution problem and the diverse issues stakeholders aim to address are increasingly complex, so too is the range of policy responses and solutions being adopted by governments, industry, non-governmental organisations (NGOs) and citizen groups. This section outlines the broadening range of policy responses and solutions, focusing on solutions pursued at the national level that are focused on product disposal and end-of-life; consumers and retailers; and producers. A small selection of these is shown in Table 1 below.

**Table 1.** Existing national initiatives—by actor and degree of enforcement.

| Focus | Voluntary | Regulated and Enforced |
|---|---|---|
| **Disposal and end-of-life focused** | - Plastic clean up initiatives<br>- Commitments to invest in waste management and recycling initiatives in developing countries | - Bans on imports of plastics wastes (China, Malaysia, Thailand, Vietnam and others)<br>- Mandated producer responsibility for single-use plastics, including deposit refunds, product take-back and recycling targets<br>- Mandatory recycled content quotas |

**Table 1.** *Cont.*

| Focus | Voluntary | Regulated and Enforced |
|---|---|---|
| **Consumer and retailer focused** | - Commitment to use less plastic packaging and single-use plastic<br>- Commitments to trial and use less plastic-intensive business models<br>- Commitments to use "greener" plastics and non-plastic substitutes<br>- Zero waste initiatives | - EU, Canada, India and Rwanda, among others, ban on retail use of single-use plastics, such as regulation on plastic bags<br>- Taxes and consumer fees on single-use, take out containers and cups<br>- Import bans on single-use plastics |
| **Producer focused** | - Commitments to increase recycled content<br>- Commitments to promote products that are recyclable and recycled<br>- Commitments to reduce plastic packaging | - Extended producer responsibility fees (e.g., France)<br>- Bans on specific plastic products, materials or production levels, e.g., of plastic bags<br>- Bans on production and use of microbeads<br>- Regulation on chemicals and chemical inputs into plastics |

Source: authors, based on annual reports, UN agreements and individual NGO and national publications.

2.3.1. A Tradition of Voluntary Initiatives

A dominant focus of efforts to combat plastic pollution has been voluntary initiatives by citizens and industry to prevent and clean up plastic waste, especially in the marine environment, and to improve plastic waste management (Clapp 2012; Clapp and Swanston 2009; Dauvergne 2018b). Numerous citizen campaigns to 'break free from plastic' and promote 'zero waste' are helping to curb some excessive, unnecessary and environmentally harmful uses of plastics. A range of companies, especially plastic producers, have committed to a variety of partnerships, such as the Alliance to End Plastic Waste, to support plastic clean up, waste management and recycling capacities in developing countries (Alliance to End Plastic Waste 2020). Alongside, a range of firms making or selling consumer products, food and beverages and toys have long supported campaigns encouraging consumers and households to better manage their waste. Increasingly, a range of major brands and retailers are making commitments to use less plastic packaging and to phase out certain single-use plastics, as well as to test and invest in less plastic-intensive business models and non-plastic business models. A range of companies have, for instance, issued targets aimed at increasing the share of recycled content in their products, as well as the recyclability and reusability of their products, and their use of biodegradable or compostable plastics. Alongside, a range of small companies and some of the world's largest retailers are innovating in the development of re-use, refill and repair retail and distribution systems that avoid plastic packaging and in-built obsolescence.

Voluntary efforts to promote a circular economy for plastics have drawn together a broad range of business and civil society groups, as well as governments. As part of its work to promote a circular plastics economy, the Ellen MacArthur Foundation's (EMF) Plastics Pact network, for instance, creates a common platform with targets to reduce unnecessary plastic use; spur innovations in design and manufacture that enable reuse, recycling, easier collection and longer life of plastic products; and that ensure plastic waste has a commercial value so that it can be recycled or "upcycled" (EMF 2019c; Packaging Insights 2020; OECD 2018b). In 2019, EMF succeeded in encouraging over 30 global brands to disclose their plastic packaging volumes as a key step toward transparency and pressure to undertake concrete measures to reduce these (EMF 2019a). On the flip-side, these 30 global brands represented only 20% of all the single packaging used globally (EMF 2019b).

Alongside valuable continuing efforts to promote disclosure and to spur companies to implement their public commitments (EMF 2021), there is broadening recognition of the challenges of accountability of the widening array of voluntary corporate pledges (Rucevska and Villarrubia-Gómez 2020). Further, some dimensions of the circular economy agenda (such as greater recycling) attract greater business interest and engagement than

others (such as the prevention of plastic pollution, reduction of plastic use and re-use of plastics), raising concern among environmental advocates that some businesses are distorting the circular economy vision and greenwashing business as usual. Some of the petrochemical and plastics producers investing in partnerships to clean up waste and improve plastic waste management face charges of hypocrisy for simultaneously working against efforts to reduce virgin plastic production and rein in the proliferation of plastics (Packaging Insights 2019).

### 2.3.2. The Move from Voluntary Initiatives to Proactive Government Engagement and Mandatory Action

Alongside the voluntary efforts of businesses, civil society organizations and citizens, governments are increasingly introducing an array of policies to support plastic pollution reduction and circular economy approaches. The shift toward mandatory action involves a combination of 'incentives' and 'sticks' deployed through a range of government policies and legislation, while also promoting voluntary action.

Examples of policies that governments have introduced include restrictions on the use of non-recyclable packaging, bans on the disposal of certain kinds of plastics, bans on the imports of certain plastic wastes, mandatory requirements for the use of a certain percentage of recycled content in products, and taxes and consumer fees on certain single-use products, such as take out plastic bags, containers and cups. Further, a growing number of governments has implemented measures that ban or restrict the import, sale, use or disposal of certain types of plastic, single-use plastics (such as plastic bags) and microbeads (European Commission 2019; Nielsen et al. 2019; Ritch et al. 2009; Excell 2019). Interestingly, developing countries are particularly active in banning the use and import of certain single-use plastics, such as bags, straws and cutlery (Barrowclough and Vivas-Eugui 2020). By July 2018, for instance, 127 countries had adopted legislation to regulate plastic bags, from outright bans in the Marshall Islands to progressive phase-outs and laws that incentivise the use of reusable bags (Excell 2019). Further, a range of governments, including the EU, Australia and India, have introduced laws that oblige producers to be financially or physically responsible for the end-of life clean-up or recycling of their products through extended producer responsibility (EPR) schemes (including deposit refund and product take-back schemes for single-use plastics (DEFRA 2019).

The adoption by a range of governments of new policies to promote more circular plastic economies reflect the emerging links being made between reducing plastics waste and the need to limit the production of unnecessary, harmful and problematic plastics in the first place.

In 2018, for instance, the European Union (EU) adopted the world's first comprehensive European Plastics Strategy in 2018 (European Commission 2018a, 2018b). It aims to raise consumer awareness and change consumer behaviour with an eye to reducing the demand (and eventually supply) of some forms of plastic, starting with single-use plastics for which substitutes are easily available and affordable. For other plastic products, the strategy aims to reduce their use through design and labelling requirements and standards, as well as waste management/clean-up obligations for producers, including through taxes and fees on some kinds of plastics. At the same time, circular economy policies also include efforts to promote investment in plastic substitutes and new business models that eliminate or reduce use of plastics, including through refill and reuse retail models.

### 2.3.3. A Growing Focus on Addressing Root Causes

Among policymakers and campaigners working to reduce plastic pollution, there is growing recognition of the need to address root causes of expanding plastic waste and pollution across the entire life-cycle. Here, a core focus is on how totackle the growing scale of plastic production and the proliferation of plastics, which in turn prompts a focus on investment in fossil fuels and virgin plastic production, and related subsidies, as well as climate policies focused on decarbonization as complementary tools.

Already, a number of governments have introduced bans not just on the import or consumption of specific plastic products but also on their manufacture linked to environmental harm, such as certain single-use plastics and polystyrene (Excell 2019), as well as bans or restrictions on certain toxic chemicals and additives used in plastics. The Break Free From Plastics movement, consisting of over 1000 NGOs and millions of citizens worldwide, calls for legislation to phase-out the production and use of many types and applications of plastic altogether. Zero waste initiatives by citizens organizations and local governments for instance, are actively counterclaims that single-use plastic packaging is necessary, such as to prevent food waste (Schweitzer et al. 2018).

Many campaigners on plastic pollution argue that the focus on recycling plastic waste diverts attention from making industry take greater responsibility for the need to stop the growing proliferation of plastics and plastic pollution. The Indonesian Zero Waste Alliance observes, for instance, that "[f]raming marine litter as only a waste management problem is nonsense when it is actually a reflection of the industry's refusal to take responsibility for the plastic pollution crises ... We can't recycle toxic plastics and pretend that the marine litter chaos is a waste issue; it's a toxic product issue" (Break Free From Plastics 2019). Among the recommendations of civil society groups campaigning on plastic pollution is for extended producer responsibility (EPR) policies that compel producers to pay costs related to handling and cleaning up toxic plastic waste.

A related concern is that the impact of plastic waste and pollution may be experienced far beyond the shores in which plastic products were produced. To tackle this offshore impact there are calls for calls for international extended producer responsibility (OECD 2018a), to put pressure on major producers and users of plastic products (including supermarkets and retailers that are expanding their presence in developing countries) to take greater responsibility not only for cleaning up waste but for reducing waste (WWF 2019).[2] Further proposals include calls for commercial plastic polluters to be made subject to civil liability claims (a precedent for such liability exists for oil pollution damage)[3] and to create a plastic super fund that could secure resources for pollution prevention and clean-up efforts. Alongside, there are calls for mandatory disclosure by companies of their plastic footprints as a baseline for reductions—an approach which UNEP has advocated (UNEP 2014) and the EMF has been advancing through their effort to promote voluntary disclosure on plastic packaging volumes (EMF 2019a).

Notably, a number of companies and industry collaborations now acknowledge the need to reduce the use of virgin plastic feedstocks in production. Leading international environmental NGOs, such as the World Wildlife Foundation, are also calling for reducing virgin plastic production (WWF 2019). On the private sector side, the Sea the Future initiative galvanised by the Minderoo Foundation has proposed that companies should pay a voluntary 'fee' on virgin plastic inputs as a way to support markets for recycled plastics and to boost incentives for recycling (Minderoo Foundation 2021). Although few commitments exist to limit expanding production of plastics, recently proposed legislation in the US Congress and some British Members of Parliament are urging a national "plastics budget" that would introduce legally binding goals to reduce plastic production. Both the UK and Italy are developing policies that apply taxes to virgin plastic inputs into plastics manufacturing.

There are also calls for the financial sector to accelerate investment in what will require a multi-billion-dollar transition of the plastics economy, demanding investment in new business models, materials and technologies (EMF 2019a). In addition, civil society campaigns are increasingly focusing on identifying and targeting major producers and users of plastics as drivers of the plastic pollution crisis (Heinrich Böll Stiftung and Break Free from Plastic 2019; Minderoo Foundation 2021), highlighting the risks associated with investment in the sector (Thoumi and Willis 2021), with some calling on institutional investors and insurers to divest from the sector (CIEL 2019; Client Earth 2018).

Building on critical analysis of fossil fuel subsidies and options for their reform (Skovgaard and Van Asselt 2018), several preliminary studies are emerging of how these

subsidies, as well as additional subsidies to the petrochemicals sector, drive the proliferation of cheap plastics (Skovgaard 2021; Steenblik 2021; Deere Birkbeck and Sugathan 2021). Following the commitment made by governments in the 2021 Glasgow Climate Pact to phase-down fossil fuels (UNFCCC 2021), pressure for the reform of subsidies to fossil fuels and the related petrochemicals and plastics sector is set to intensify. The growing attention to root causes, and to links between the climate and plastics crisis, is also evident in calls for plastics to be a stronger focus of EU climate policies, including its efforts to drive down emissions through the EU's new Green Deal and the European Commission's proposal for a Carbon Border Adjustment Mechanism (CBAM) (Ruggiero 2021).

### 2.4. An Emerging Focus on Development Issues

A new angle of interest relates to the environmental and economic challenges associated with plastic pollution in developing countries (Ettinger 2015; Van de Klundert and Lardinois 2017; Schröder 2017, 2018). Many developing countries are actively involved in efforts to improve plastic waste management, reduce unnecessary use of plastics and promote more circular plastic economies. As noted above, concern about plastic pollution is spurring developing countries to deploy trade restrictions and bans on certain specific single-use plastic items and plastic wastes. However, systematic study of the development constraints developing countries face in reducing plastic pollution and transforming their own plastics sector and use of plastics is just beginning (Vince and Hardesty 2017, 2018; Schröder 2018, 2020; Williams et al. 2019).

A key consideration is that developing countries are projected to increase their use of plastic in the coming decade, as rising incomes increase the demand for packaged goods and new products. Cheap plastic has pushed out traditional products in local markets, boosted by global value chains and imported business models that rely heavily on plastic products and packaging. Inadequate provision of fresh water means millions of people in developing countries rely on water bought in plastic bottles. Some developing countries have also become major producers of primary plastics as well as manufacturers of plastic products—for both domestic and international markets—and also have economic and employment interests tied to incineration and recycling businesses.

From a political economy perspective, there are also important special interests at hand. Plastic waste brokers and traders, for instance, have significant commercial stakes in the plastic waste trade and may resist efforts at regulation. Many developing countries have significant informal economic activity linked to the collection, sorting, re-use and management of plastic waste. Although local populations dealing with collection and treatment of plastic can suffer signicant health impacts (Dias 2016; Gutberlet 2019), local waste pickers may resist modern waste management practices that threaten their livelihoods. In addition, retailers in many local markets in developing countries have become dependent on plastic to sell their produce and for consumers to transport it home.

The recent experience of India is instructive of the complex political economy of plastic pollution reduction efforts. In 2019, legislation to reduce consumption of plastic bags and single-use plastics stumbled under citizen and business resistance. With four million people estimated to be employed in the plastics sector in India, the proposed change was untenable economically and politically (Phartiyal and Jadhav 2018; BBC 2019; Staub 2019).

Another important factor relates to the role of plastic in the industrialisation and export strategies of developing countries. Trade in plastics is a global business worth at least USD 1 trillion annually (Barrowclough et al. 2020). Using UN Comtrade data, Figure 1 shows the authors' findings on the value of global exports of different kinds of plastic products and inputs categorised according to their place in the life-cycle of plastic.

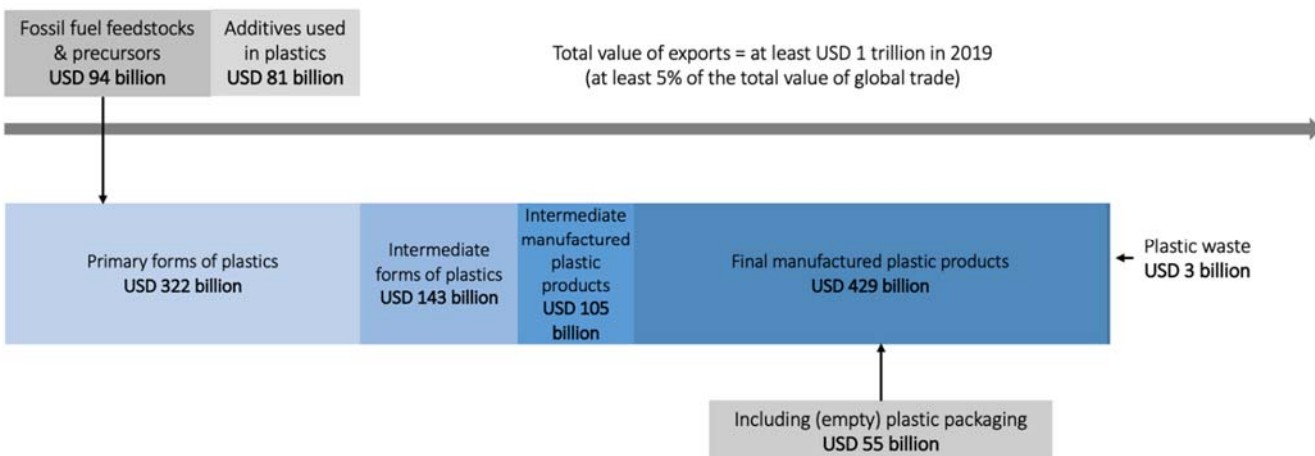

**Figure 1.** International trade across the life-cycle of plastics (exports 2019).

Virtually all developing countries are involved somehow in the import or export of plastics at different points along the life cycle, and many are engaged in both. A number of developing countries, including those actively working to reduce plastic pollution, are heavily invested in the export of plastics to regional or international markets, and for many this is a critical export sector and one they hope to consolidate and expand. Indeed, plastics are an essential input into many products and processes linked through global value chains that developing countries see as the path to higher incomes and industrial diversification (Figure 1). Even non-industrialised countries relying on agricultural production find that the move to higher value-added exports relies on plastic packaging to comply with regulations in export markets, such as phytosanitary rules. In addition, over 25 developing countries derive an important share of their export revenues from fossil fuels (sometimes as much as 90%), which are increasingly destined for plastics production (IEA 2020). Given the significance of trade in plastics and the trade-related challenges to transformation in developing countries, careful consideration of appropriate international trade policy will be an important part of the policy tool kit for dealing with the upstream challenges of plastic pollution (Deere Birkbeck 2020a, 2020b; Deere Birkbeck and Sugathan 2021; OECD 2018c) (discussed in Section 4).

Finally, given the valuable export markets that exist throughout the life-cycle of plastic products, as illustrated in Figure 1, developing countries are increasingly alert to the potential to expand their production, use and export of natural fibre-based substitutes for plastics to meet some of the needs of plastic consumers, using inputs such as jute, abaca, coir, kenaf, sisal (known collectively as JACKS) and others (UNCTAD 2021b). Products used such natural fibres may not necessarily replace all plastics use but sustainably produced substitutes can be used strategically, especially for applications where some of the properties of plastic are dispensable (Material Economics 2018b; UNCTAD 2021b). Some countries already have a comparative advantage in their production and are interested in exploring options to support and foster local business start-ups to produce alternatives to plastic packaging and single-use plastics made from JACKS and bamboo for domestic markets (and thereby replace certain plastic imports) as well as for export markets (Excell 2019). While studies of the potential are still underway (UNCTAD 2021b), it is notable that some developed countries, such as the UK, are including support for plastic substitutes in their overseas development assistance activities, alongside other activities to reduce plastic pollution.

## 3. Evolution of the International Policy Landscape

Informed by evolving understandings of the nature of the plastic crisis, its causes and potential solutions, the international policy landscape is also evolving. At the international level, there is a broad and diverse range of policy initiatives and inter-govermental processes

that aim to build on, coordinate and expand voluntary efforts and government approaches discussed in Section 2. Amidst the search for strategies and policies that cannot only reduce plastic pollution but also transform the plastic industry to better meet the climate sensitive needs of the 21st century, the current state of play in international policy discussions is extremely fluid with numerous proposals and pathways being discussed and pursued with varying levels of intensity.

This section briefly reviews the landscape of international policy initiatives that have emerged in three overlapping phases, relating to (a) a focus on plastic pollution in the marine environment, (b) a broader focus on plastic pollution across the life-cycle, including its health and development impacts, and (c) an incipient focus on plastic production and trade. A sub-set of these initiatives is shown in Table 2.

**Table 2.** Evolution of the international policy environment.

| Phase 1: Evolving Policy Frameworks on Marine Litter and How to Better Manage Waste | Phase 2: Intensified and Wider Focus on Marine Plastic Pollution | Phase 3: Widening Focus on Plastic Pollution Across Life-Cycle, Concerns about Plastic Production and Action on Plastic Waste Trade |
|---|---|---|
| Including: 1969 Creation of Joint Group of Experts on Scientific Aspects of Marine Environment 1972 London Convention on the Prevention of Marine Pollution by Dumping of Wastes and Other Matter 1973 International Convention for the Prevention of Pollution from Ships (MARPOL) 1982 UN Law of the Sea includes legal requirements to "prevent, reduce and control pollution of the marine environment from any source" 1995 UN Global Programme of Action for the Protection of the Marine Enviroment from Land-based Activities (GPA) 2004 UN General Assembly resolution 52/24 creates open-ended Informal Consultative Process on Oceans and the Law of the Sea, recommending a focus on marine debris 2012 UN Global Partnership on Marine Litter (GPML) | Including: 2014 UN Environment Assemby Resolution on marine plastic debris and microplastics 2015 Targets related to marine pollution feature in the UN Sustainable Development Goals (SDGs), including Target 14.1 2015 Implementation plan for achieving 2020 goals for sound chemical management approved at the fourth session of the International Conference on Chemical Management (ICCM4) including legal frameworks that address the life cycle of chemicals and waste 2016 UNEA resolution on marine plastics and microplastics 2016 UN Open-ended Informal Consultative Process on Oceans and the Law of the Sea focuses on marine debris, plastics and microplastics | Including: 2017 Ocean Conference declaration includes commitment to "develop sustainable consumption and production patterns" 2018 G7 Oceans Plastic Charter commits action to lifecycle management approach to plastics 2019 Basel Convention plastic waste amendments and creation of Plastic Waste Partnership 2021 WTO Ministerial Statement on Plastic Pollution and Environmentally Sustainable Plastics Trade 2021 Statement of the Ministerial Conference on Marine Litter and Plastic Pollution [2022 Potential UNEA decision to launch negotiations for a Global Agreement on Plastic Pollution] |

Beginning in the 1960s, Phase 1 of international engagement on plastic pollution lasted over 40 years and focused on the impact of plastic debris on the marine environment. This first Phase saw the emergence of a number of international frameworks to reduce marine pollution that are still relevant today, including government declarations to reduce marine debris, the conclusion of the Law of the Seas, and the creation of several United Nations partnerships, expert bodies and consultative processes to address marine debris and litter.

Between 2012 and 2014, inter-governmental cooperation entered a shorter but important Phase 2, where the focus on marine plastic pollution expanded to incorporate attention to the scale and impacts of microplastic pollution. This phase was also marked by a pronounced emphasis on the need to address the land-based sources of marine plastic pollution, including inadequate waste management (Ocean Conservancy and McKinsey 2017). During this phase, there was also growing attention to challenges of poor waste management for a range of materials, including the associated impacts on developing countries. The UN Sustainable Development Goals emphasized the need to promote both more sustainable production and consumption, and also more effective waste management systems

and infrastructure in developing countries. Alongside, governments also intensified their efforts to improve international cooperation on sound management of chemicals through UN processes.

Only in Phase 3, starting around 2017, did the need to address the "upstream" production of plastic start to attract more focused attention on the international policy agenda. The June 2017 UN Declaration "Our Ocean, Our Future: Call for Action" included 178 intergovernmental commitments to act on marine plastic pollution, one of which cites the need to "develop sustainable consumption and production patterns" (UN 2017). Further, in 2018, an ad hoc open-ended expert group created by the UN Environment Assembly emphasised the need for a more integrated approach to tackling plastic pollution across the entire life-cycle of plastics, from production to disposal (UNEP 2018b). Among other suggestions, the expert group called on governments to use incentives to foster more sustainable production and consumption and requested guidelines on plastic use and production, including information on standards and labels to incentivise consumers and businesses to adopt sustainable products and practices (UNEP 2018b).

Amidst growing evidence of plastic waste trade to countries without capacity for environmentally sound management, governments have taken more decisive action to strengthen relevant international law. By early 2019, over 180 countries (with the notable exception of the United States) had agreed to amend the Basel Convention to regulate and improve transparency of plastic waste trade, including through requirements for prior informed consent for the shipment of certain plastic waste, and formed a Plastic Waste Partnership to share technical skills, best practice and financing. Alongside calls from the environmental community for swifter and more stringent implementation of the Basel 'plastic waste amendments', the impacts of plastic waste import restrictions on waste generation and collection as well as global markets for recycling and recycled plastic materials are attracting growing interest (Brook et al. 2018; Staub 2017; Uhm 2021; WEF 2020).

A key feature of international policymaking on plastic pollution in phase 3 is the growing recognition that tackling this crisis will require a "system change" approach not only nationally but also at the international level (The Pew Charitable Trusts and SYSTEMIQ 2020), and that this will demand greater attention to the right mix of economic regulations and policies to mandate and incentivize action and innovation. Further, there is growing consensus among governments and a diversity of stakeholders on the need to move from a patchwork of voluntary and ad hoc initiatives to a binding and coordinated international framework for reducing not only marine plastic pollution but plastic pollution across the life-cycle of plastics (Villarrubia-Gómeza et al. 2018; Simon et al. 2021) (discussed in Section 4).

## 4. Where Now? Strategic Debates and New Directions for Global Governance of Plastics and Plastic Pollution

This section takes stock of strategic debates on the global governance of plastics and plastic pollution and new directions for international cooperation. While the focal point of most policy action is international environmental organisations and processes, complementary discussions are emerging in the realms of international economic policy, especially in areas of trade and investment. These new directions are not yet well covered in the broader academic literature as they are new and fluid, and in some instances, still exploratory. The aim of this section is to take stock of some of the state of play and to highlight research gaps for the future.

### 4.1. International Environmental Cooperation

Work on different aspects of plastic pollution is underway in numerous international environmental organisations, processes, and negotiations, reflecting different views of the diversity of relevant policymakers, environmental experts and stakeholders views on key priorities, as well as on the most efficient and effective ways forward (UNEP 2018a;

UN EMG 2021). A unifying thread across these for an over the past five years has been recognition that stronger international cooperation on plastic pollution is needed.

One approach has been to improve ocean governance frameworks, including through better use of existing international instruments (such as the United Nations Convention on Law of the Sea (UNCLOS) (Ocean Plastic Legal Initiative 2018). Alongside, there have proposals to integrate plastic pollution priorities into negotiations launched in 2017 for the protection of biodiversity in areas beyond national jurisdiction (BBNJ) (Tiller and Nyman 2018). Following the successful conclusion of negotiations on the 2019 'plastic waste amendments' to the Basel Convention, work is now underway advance their effective implementation, explore options for further amendments, and to support work related to the Convention's Plastic Waste Partnership and its work aimed at environmentally sound waste prevention and management. Alongside, governments and stakeholders are exploring options under the Stockholm Convention to eliminate or restrict the production and use of persistent organic pollutants used in plastics.

Meanwhile, there are growing calls from governments, civil society groups and a range of businesses for the launch of negotiations on a new global environmental agreement to tackle plastic pollution (Haward 2018; Simon and Schulte 2017; Raubenheimer and McIlgorm 2017, 2018; Raubenheimer and Urho 2020; RECIEL 2018; Tiller and Nyman 2018; Carlini and Kleine 2018; CIEL 2019, 2020; Simon et al. 2021; WWF et al. 2020). Despite the array of voluntary international initiatives and national efforts, proponents of a global treaty highlight significant gaps in national and regional legislation and commitments; the absence of global rules, standards and recommended practices for reducing plastic pollution, and the lack of global obligations to prevent plastic pollution as well as common reporting and monitoring requirements; and emphasize the need for greater international coordination, including commitments to relevant financial and technical support for states (WWF et al. 2020).

For some, the core motivation and purpose of a new global environmental agreement should be to consolidate international cooperation on marine plastic pollution, with a primary focus on clean-up and remediation efforts, improved waste management to reduce land-based and sea-based sources of pollution, along with incentives for recycling and innovation to promote more recyclable plastics. Alongside, there are proposals for a treaty that caps production of certain kinds of plastic. Here, scholars have explored the possibilities for building a treaty modelled on the successful Montreal Protocol on ozone-depleting substances (that aimed to reduce and replace use of chlorofluorocarbons (CFCs) (Raubenheimer and McIlgorm 2017). Further, the Break Free From Plastics movement argues that the proposed global agreement should stop the development of new petrochemical and plastics production infrastructure (CIEL 2019, 2020; WWF et al. 2020).

In 2020, the Nordic Council of Environment Ministers commissioned work on possible elements of a new global agreement to tackle marine plastic pollution (Raubenheimer and Urho 2020), which highlighted among other elements the importance of cooperation on international standards across the life-cycle of plastics (Raubenheimer and Urho 2020). A joint civil society-business report on the 'business case' for a new global treaty argued that: "global goals and binding targets, together with national action plans and consistent measurement is needed to harmonize policy efforts, enhance investment planning, stimulate innovation and coordinate infrastructure development" (WWF et al. 2020). It concluded that "while voluntary initiatives can deliver change among market leaders, an international binding approach is needed to deliver the necessary industry scale change" (WWF et al. 2020).

Since mid-2020, a number of governments and advocacy groups have galvanised behind the idea that any new global agreement should address plastic pollution across the whole plastics life-cycle (Simon et al. 2021). As of late 2021, over 80 governments had endorsed a ministerial statement in favour of a new global agreement to tackle plastic pollution. In that statement, governments noted that: "Owing to the nature of global supply and value chains, trade in plastic waste and the flow of plastic in the ocean, the

challenge of plastic pollution and marine litter is transboundary and global in scope. Current approaches, which are limited geographically and consider only parts of the life cycle of plastics, have proven insufficient. They cannot address the scale of this challenge, let alone keep pace with predicted future developments. Consequently, the time has come for countries and stakeholders to ramp up their efforts and take collective, balanced, ambitious and decisive action."[4]

In the lead up to the UN Environment Assembly (UNEA) in early 2022, two proposals for UNEA resolutions on the launch of negotiations have been tabled by governments, one led by Rwanda and Peru proposal, which reflects the life-cycle approach taken in the ministerial statement, and another by Japan, which focuses more tightly on marine plastic pollution and addressing problems of waste management and prevention (CIEL and EIA 2021).[5] Meanwhile, within the UN system, there is growing attention to the need for more coordinated, integrated, and system-wide approaches that galvanise the collective contribution of many parts of the UN system and multiple stakeholders as vital to addressing not only marine plastic pollution (UN EMG 2021) but also plastic pollution across the life-cycle of plastics, including the climate footprint of the plastics economy.

*4.2. Complementary Approaches to Reducing the Proliferation of Unnecessary, Harmful and Problematic Plastics at Source*

Pulling out of unnecessary, harmful and problematic use of plastics will require a multi-faceted and integrated approach that addresses all actors and stakeholders in the global plastics economy. This in turn will require approaches that draw together a range of different international actors, including the range of relevant international actors from the across the UN system—from those working on marine plastic pollution, chemicals and climate change to those working on waste management, urban infrastructure, finance and trade, and in sectors ranging from tourism and agriculture to fisheries, consumer goods and retail.

An important next step is to situate the plastic problem within the wider political economy of plastic production, trade and consumption, with a stronger focus on the role that economic policies and tools can play as a part of, and complement to, efforts to strengthen international cooperation to reduce pollution across the plastic life-cycle. To make progress, attention to the important downstream dimensions of plastic pollution described above must be complemented by efforts to address underlying "upstream" causes of plastic pollution across the life cycle and support structural change required across that life-cycle.

Looking ahead, reducing plastic pollution will require the development and implementation of integrated policy frameworks that can support both sustainable transformation of the global plastics economy and a just transition. Table 3 below identifies some of the range of policy tools required in an integrated international policy framework to achieve those dual ends in both developed and developing countries.

Critically, the economic constraints and trade-related challenges ahead may differ in developing countries from those in more advanced economies, or where the plastics industry has been invested longer. There is particular need for attention to the financial and industrial policy and institutional environment necessary for the growth of innovative and environmentally friendly "sunrise" industries (UNCTAD 2017; Barrowclough and Kozul-Wright 2017; Barrowclough and Vivas-Eugui 2020) that reduce the unnecessary, harmful and problematic production and use of plastic, and in particular plastic waste, while generating new economic opportunities for developing countries.

Without wishing to understate the scale of the challenge ahead, much can be learned from the use of industrial policies to advance successful transformations in other sectors. There is no case in history, for instance, where a country has achieved the structural transformation of moving from subsistence agriculture to industrialisation without the use of such industrial policies (Amsden 2001; UNCTAD 2017; Haldane 2018; Mazzucato 2011). Today, industrial policies are still widely used by many countries and have been underpinned

by the move to revolutionary new technologies, such as solar power, the Internet and the iPhone (Mazzucato 2011).[6]

Encouragingly, some developing countries have comparative advantages in the production of non-plastic substitutes, such as products and packaging based on natural fibres that can be produced locally (UNCTAD 2021b). While these are viewed as a development opportunity, developing country businesses working on plastic substitutes and business models that use less plastic face many challenges and obstacles compared to the large and well-known businesses, which enjoy access to plentiful cheap finance; they may also face intellectual property and other barriers that limit affordable access to relevant technologies and will require support for research and investments to ensure new technologies and business models are environmentally sound. Developing countries already invested in the petrochemicals and plastics sector will also be especially reliant on industrial policies that can support effort to decarbonize these sectors, shift toward less polluting alternative plastics, and promote more circularity, including through a phase out of some plastics production.

**Table 3.** Integrated policy framework for sustainable transformation of the global plastics economy and a just transition.

| Promoting Sustainable Transformation Across the Life-Cycle of the Plastics | Ensuring a Just Transition to Support the Process of Transformation |
|---|---|
| - **Policies, rules and regulations** to require and enforce more sustainable production (including taxes, charges and extended producer responsibility).<br>- **Trade policies** to support national efforts to reduce unsustainable plastics production and consumption and to encourage alternatives—including targets and commitments to reduce trade in certain plastics; boost trade in environmentally sound waste management technologies and services; and promote substitute products and delivery modes.<br>- **Correct pricing** for plastics and internalize costs of environmental impacts, including through **disciplines on subsidies** that sustain/boost harmful production.<br>- **Financial and industrial policy levers** to give incentives for industry to adapt production, manufacturing and retail models to reduce plastic use, shift to more sustainable alternative plastics and non-plastics substitutes.<br>- **Support multilateral and regional development banks** and institutions to finance transformative leaps away from unsustainable plastics by firms and investors;<br>- Incentives and disincentives related to ownership of technologies and related IP that can support efforts to reduce plastic pollution and decarbonize production;<br>- **Incentives for producers**, manufacturers and retailers to adapt existing process and products; and<br>- Boosted **demand and supply of substitutes and new business models** through procurement policies at national and regional level.<br>- **Develop and use sustainability standards** for plastic products and production methods, including label and certification of environmental standards. | - **Support for research**, technical assistance and Aid for Trade to support developing countries active in GVCs involving plastics.<br>- **Technology transfer** for developing countries to adapt existing methods and introduce new ones (consideration of ownership of alternative and substitute technologies as well as opportunities for MSMEs from developing countries needed).<br>- **Capacity building** and support for domestic production and trade in waste management services and technologies.<br>- Clear **sun-set periods** for removal of existing incentives for the production of unnecessary, harmful and problematic plastics.<br>- **Social policies, including incomes support for temporarily** displaced workers, social services for permanently displaced works, and transitionary support for removal of subsidies.<br>- R&D, education and skills policies for retraining in the use of new processes and products.<br>- Cooperation among international organisations and processes.<br>- **Information exchange**, monitoring and assessment (e.g., on trade-related measures on plastic pollution). |

Transforming the global plastic economy will require governments to deploy a range of industrial policies. Governments will need to promote research and development in product design that reduces overall plastic use and in environmentally sound waste management technologies, alternatives plastics, non-plastic substitutes, as well as in strategies to decarbonize the sector—all of which are unlikely to emerge from the market without

enhanced government support or incentives. Government policies related to intellectual property rights and licensing will also need to support businesses to secure affordable access to technologies vital to the shift to alternative plastics and non-plastic substitutes.

Financial policies will also be needed to help create capital and guide it to new uses that reduce plastic pollution and promote a more circular plastics economy. In terms of public finance, governments must devise strategies to phase out subsidies that sustain, and drive expansion of, the fossil fuel and petrochemical sectors, lowering the costs of plastic production and flooding the world with cheap plastics (Deere Birkbeck and Sugathan 2021). At the same time, public finance will need to be repurposed toward pollution reduction goals. Significant finance will be needed to provide incentives for plastics producers, product manufacturers, brands and retailers—from large established players to new start-up firms and small businesses—to take the leap involved to innovate, moving away from tried and tested markets toward new markets that may be unknown and risky.

Promoting a move away from plastics needs an articulated and supportive system ideally starting at the apex, with central banks, and filtering down to specialist banks that can finance loans and investments to firms that can drive the transformation of the plastics sector. Since the 2007–2008 global economic crisis and even more since the COVID-19 crisis, central banks have already shown an increasing willingness to create credit and guide it to desired directions. In many countries, for instance, central banks have already been re-evaluating their operations and mandates in light of the climate crisis and the need for decarbonization (e.g., Bank of England 2021; Campiglio et al. 2017; Carney 2015; Dikau et al. 2020; UNCTAD 2019). Looking ahead, their strategies with respect to decarbonization will need to address the interlinkage of the fossil fuels and plastics industries, ensuring that pressure to move out of fossil fuel-based energy does not drive a shift to ever more fossil-fuel driven production of unnecessary, harmful and problematic plastics (Bauer and Nielsen 2021). Unfortunately, recent analysis shows that COVID-19 relief and recovery policies in a broad range of companies have been privileging the very large and long-standing fossil fuel and petro-chemical companies that drive plastic production (Barrowclough and Finkill 2021; OilChange International 2021; Tearfund 2021).

Development banks also have an important role to play in supporting transformation, including regional development banks that pool finances and help to lend money for inter-regional projects and reduce costs of transition for individual countries. In Europe, for instance, the European Investment Bank (the EU's regional bank) is already providing financial and advisory support towards the transition to a circular economy, which could also be harnessed for the plastics economy. At the same time, many development banks currently actively support (in the form of loans, guarantees, and export credits) that subsidize the expansion of capacity for new petrochemical and plastic production in developing countries (Deere Birkbeck and Sugathan 2021; Steenblik 2021; Skovgaard 2021). Just as multilateral development banks face pressure to end support for fossil fuel expansion in developing countries, they can expect pressure to critically re-evaluate and phase out support that will further lock in expanding production of cheap conventional plastics.

Credit rating firms will also need to pave the way forward. Already in 2019, a leading international credit rating firm observed that the credit ratings of European packaging firms were under threat due to public concerns about plastic packaging (Moody's Investor Service 2019). A related, supportive move will be required from insurance companies. Amidst intensifying scrutiny of major publicly-listed companies responsible for the production and use of single-use plastics (Minderoo Foundation 2021) and the push for stronger international regulation, financial analysts are signalling the reputational and financial risk to insurers as well as investors and companies from over-exposure to plastics (Client Earth 2018; Thoumi and Willis 2021; UNEP 2019).

To pursue the array of industrial, financial and other policies required for transformation and just transition, governments will need to have sufficient national policy space to tailor and implement such policies, which will rely on a supportive global policy framework. International trade, finance and investment policies will need to crafted in ways

that enabling and support a sustainable transition to a decarbonized circular economy that produces and uses less plastic, and that supports developing countries to adapt, pursue and advance development goals in this context.

There is no one venue for addressing the trade, finance and technology dimensions of the global plastics economy and plastic pollution. Over the past two years, interest in the intersection of environmental and international economic policymaking and options for more integrated approaches to tackling plastic pollution informed by development priorities is already intensifying, spurred in part by advocacy for a decarbonised and more circular global economy.

Over the past several years, the trade dimension of the plastic pollution crisis has spurred not only the Basel Convention's plastic waste amendments but also growing interest in the relevance of trade and trade policies to plastic pollution and their potential role in supporting solutions (Deere Birkbeck and Sugathan 2021; Vivas-Eugui and Pacini 2021; WEF 2020). At the national level, many countries are already implementing a range of trade measures and policies to support plastic production and exports (Deere Birkbeck et al. 2021). At the same time, as noted above, a growing number of countries are adopting national plastic pollution policies that have a trade dimension, including trade bans and restrictions, as well as a range of standards and requirements for plastic products plastics and packaging that are also relevant to trade and international supply chains (Deere Birkbeck et al. 2021). Presently, however, there are important gaps in transparency, coordination and coherence of policies on trade and plastic pollution. Some governments face bilateral trade pressures that threaten their national plastic pollution reduction efforts; asymmetric trade negotiations that push countries to accept expanded imports of plastic waste and products are a case in point (Deere Birkbeck and Sugathan 2021). Some countries also export plastic products and chemical additives that are restricted in domestic markets (OHCHR 2021).

A growing range of international economic organisations are engaged on issues of plastic pollution, circular economy and trade, and are a site for intergovernmental discussion of these issues, including the WTO (2018, 2021), UNCTAD (2020, 2021a) and the OECD (2018b) and the International Organization for Standardization (ISO) (Weissinger 2021). Bans on imports of non-industrial plastic waste have been discussed in several WTO committees (WTO 2018) and in late 2020, a group of WTO members launched the Informal Dialogue on Plastic Pollution and Environmentally Sustainable Plastics Trade.[7] In June 2021, trade ministers of 65+ WTO Members issued a ministerial statement at the WTO outlining their shared commitment to exploring options to use trade policy to support other international efforts to reduce plastic pollution (WTO 2021). Among their goals are to boost publicly accessible data, monitoring and analysis of trends in global trade flows across the life-cycle of plastics and plastic supply chains, as well as their implications for the design of trade policy measures to reduce plastic pollution. In addition to dialogue on national approaches to harnessing trade and trade policies to reduce plastic pollution, the governments committed to dedicated discussion on how trade-related cooperation can help reduce unnecessary or harmful plastics and plastic products, including single use plastics and plastic packaging associated with international trade, and on how to promote trade in goods, services and use of technologies that can reduce plastic pollution (WTO 2021).

In 2021, governments also gave UNCTAD a stronger mandate on plastic pollution, calling on it to "address the discharge of plastic litter and other waste in oceans and significantly reducing marine pollution of all kinds," which will include attention to trade, finance and development issues (UNCTAD 2021a). As governments move forward with proposals to launch negotiations on a new global agreement on plastics, there is increased interest in how such a framework can incorporate attention to the trade and finance dimensions of reducing plastic pollution and transforming the plastic economy (Deere Birkbeck and Sugathan 2021; CIEL and EIA 2021; Vivas-Eugui and Pacini 2021). Possible options include cooperation to establish international targets and standards across the plastics life-cycle that can move the plastic sector and trade flows in the right direction (Deere Birkbeck and Sugathan 2021). Trade provisions that could be considered in a

proposed UNEA agreement include commitments to reducing trade in unnecessary, toxic problematic and harmful plastics (Deere Birkbeck and Sugathan 2021; CIEL 2020).

## 5. Conclusions, Future Research Needs and Gaps

This article has highlighted the need to complement important existing efforts to tackle plastic pollution through strengthened international environmental cooperation with greater understanding of the political economy dimensions and challenges at hand. In so doing it has underlined that the evolving understanding and framing of the "plastics" crisis as one that is centrally concerned with marine plastic pollution has expanded to incorporate concerns for a wider set of environmental, climate, health and development challenges related to plastic pollution across the entire plastic life-cycle. This requires focused attention to the range of international economic tools and strategies needed to transform the global plastics economy and secure a just transition, and how these can be incorporated into and complement efforts to develop a new global agreement on plastic pollution.

Looking ahead, advancing an integrated system change approach to plastic pollution at the international policy level will require attention to a number of research gaps, including through systematic study of:

- the global political economy of plastic production and the factors enabling its expansion, as well as the regulatory behaviour of key commercial actors. A better understanding of the political economy dynamics of the plastic industry and global supply chains—market structure and concentration, location of production, investment and trade flows and employment—will help identify effective solutions.
- industrial policies that can spur the structural transformation needed to stop the proliferation of unnecessary, harmful, toxic and problematic plastics; supports global commitments to phase-out fossil fuels; and support the transition to more sustainable alternatives and substitutes. Promoting change demands attention to technical, socio-economic and institutional aspects of structural transformation, and to multiple economic sectors and stakeholder groups. It will also require focused attention to the needs of developing countries, where the economic constraints and political challenges may be different from those in more advanced countries.
- the international economic regulatory environment and international policy frameworks relevant to the production of plastics and their substitutes, alongside hard and soft law instruments for international environmental cooperation.

By producing new evidence and solutions, work on this research agenda would help advance international cooperation that tackles the upstream and downstream challenges of reducing plastic pollution across the life-cycle.

**Author Contributions:** D.B. and C.D.B. have read and agreed to the published version of the manuscript. We acknowledge with gratitude the many helpful comments and insights received from participants at the international conferences, webinars and other events at which elements of this work have been presented over the years 2019–2021. Any mistakes or omissions remain the authors' own. All authors have read and agreed to the published version of the manuscript.

**Funding:** The authors note with gratitude financial support from the Swiss Network of International Studies SNIS and in-kind support from UNCTAD.

**Institutional Review Board Statement:** Not applicable.

**Informed Consent Statement:** Not applicable.

**Data Availability Statement:** Data relating to the plastics trade life-cycle database can be found on the UNCTAD website at https://unctad.org/statistics.

**Conflicts of Interest:** The authors declare no conflict of interest.

## Notes

[1]   See the Statement of the Ministerial Conference on Marine Litter and Plastic Pollution, held in Geneva, 1 and 2 September 2021, available online: https://ministerialconferenceonmarinelitter.com (accessed on 15 June 2021).

[2]   The call for international extended producer responsibility was discussed (but not adopted) in the context of the Global Partnership on Marine Litter's 2011 Honolulu Strategy and has arisen in numerous dsicussions in the context of the United Nations Environment Assembly (UNEA) context, including in proposals for a global agreement on plastic pollution.

[3]   See the CLC Convention on Civil Liability for Oil Pollution Damage, which entered into force in 1975.

[4]   See note 1.

[5]   See the draft resolution on "Internationally legally binding instrument on plastic pollution," for UN Environment Assembly-5.2 issued in 2021 by the governments of Peru and Rwanda, co-sponsored by Chile, Costa Rica, Ecuador, European Union and its Member States, Colombia, Guinea, Kenya, Madagascar, Norway, Philippines, Senegal, Switzerland, United Kingdom, and Uganda. Available online: https://wedocs.unep.org/bitstream/handle/20.500.11822/37395/UNEA5.2%20Global_Agreement_Explanatory%20note%20and%20Resolution%2027%20October.pdf?sequence=1&isAllowed=y (accessed on 15 June 2021). In June 2021, Japan also issued a draft resolution for UNEA-5.2 for an "International legally binding instrument on marine plastic pollution," which focuses more narrowly on marine plastic pollution, available online: https://wedocs.unep.org/bitstream/handle/20.500.11822/37625/Draft%20Resolution%20on%20an%20international%20legally%20binding%20instrument%20on%20marine%20plastic%20pollution_Japan.pdf?sequence=1&isAllowed=y (accessed on 5 June 2021). For a comparison of the two proposed Plaresolutions, see CIEL and EIA (2021).

[6]   In this view, a move away from dependence on plastics and towards the creation and use of different kinds of products or processes is an example of the process of "creative destruction" identified by Schumpeter as characteristic of transformation.

[7]   Notably, exports of domestically prohibited goods, in particular hazardous waste, have long been a subject of discussion at the WTO and its predecessor, the GATT, as developing countries sought to limit "dumping" of toxic wastes.

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
