# Peer review of "Transforming the Global Plastics Economy: The Role of Economic Policies in the Global Governance of Plastic Pollution"

_socsci, doi:10.3390/socsci11010026_

Round 1

Reviewer 1 Report

Summary

This manuscript summarises the progression of plastic pollution discourses and of policies to address the issue of plastic pollution by different stakeholders. Multiple examples are provided throughout the manuscript, however, this manuscript is not written in the style of an academic publication, and as such I do not feel that it can be accepted. Please find below general comments, followed by more specific comments directed to each section.

General comments

  • My main issue with this manuscript is that there is no clear research question. The entirety of section two reads as list of examples, and the manuscript more broadly lacks context. This manuscript is not a research article, nor is it a review. The summary presented makes some valid points, however, its contribution to the literature and the justification of not just its conclusions but also many of its statements, are not clear. Many examples are provided throughout the manuscript, but each section reads as an (often not comprehensive) summary of these examples.
  • The manuscript assumes too much knowledge throughout – concepts, stakeholders, and problems are detailed without any introduction, and the plastic pollution problem is not defined at all. In some sections, too few examples are used, and there is a lack of balance in the examples used throughout.
  • There is no clear identification of the work’s contribution to the field, and no coherent progression throughout the manuscript from introduction to conclusions. This manuscript presents a lot of information, but does not synthesise this information into a coherent piece of academic writing.
  • Writing is sometimes informal (e.g. ‘on the private sector side…’, line 285)
  • There are typographical errors throughout that need addressing
  • The fundamental flaw of this manuscript is the way it uses the information it presents. I hope that greater engagement from, and / or more guidance by, senior members of the authorship team (or the inclusion of senior collaborators in the team if this manuscript is authored by solely Early Career Researchers) will improve the style and structure of the manuscript should the author team wish to pursue the publication of this manuscript.

Specific comments

Abstract

  • Line 5: The abstract should be a summary of the manuscript, and its context. Questions in the abstract are not ideal, and posing a question at the start of the manuscript assumes an understanding of the context that is missing from the manuscript.

Section 1: Introduction

  • Line 25: The purpose of the paper is introduced too early. There is an assumed novelty in this framing. The authors need to define the problem and identify the knowledge gaps addressed before outlining the aim(s) and objective(s) of the work – which are not included at all.
  • Lines 25-51: This entire section reads as an extension of the abstract, stating what the manuscript does, and not outlining the topic it addresses. It does not follow the author guidelines for the introduction, which require the manuscript to ‘place the study in a broad context and highlight why it is important’. It also makes no reference to the literature and so is unable to carefully review the current state of the research field, and does not outline the work’s aim or conclusions.

Section 2: Evolving definitions of the ‘problem’ and solutions

  • Line 53: Grouping definitions with solutions does not seem logical. A ‘middle bit’ is needed
  • Box 1: This box is poorly formatted and in need of revision
  • Line 54: The problem of plastic pollution, and either its current or initial framing, is needed before its reframing can be introduced.
  • Lines 57/Box 1: The layers described are arbitrarily defined by the authors with no justification or reasoning. This currently reads as an entirely subjective framing of the issue. Why does each layer contain the information it contains? And what is the benefit of separating the evolution of perceptions and understanding into these layers? This approach needs some introduction, and careful consideration and explanation.
  • Lines 55-69: This paragraph lacks focus. It reads as a series of standalone statements that do not aid the reader in understanding the framing of plastic pollution.
  • Line 63: Define ‘microplastics’
  • Line 114: Quotation not closed and the reference to a Minsky Moment assumes niche economic knowledge that should be defined for the environmental readership this topic attracts.
  • There is no coherent argument throughout this section. Each paragraph makes points that lack context and integration into a progressive discussion.
  • Figure 1 is very poorly formatted. It is not clear what it is meant to show and the text is, in parts, illegible. It is also not referenced in the text.
  • Subheadings within sections 2.2 and 2.3 are not clear
  • Lines 196-202 and 210-212: There is a lack of referencing for several statements throughout the manuscript. These lines are prime examples of where reference to the literature is needed.
  • Section 2.3: This section in particular reads as a list of examples. These examples are not used to support a clear point that benefits the field, nor are they critically discussed.
  • Lines 223-226: With reference to just two examples, the section in lines 209-226 is not enough to support this concluding statement.
  • Table 1: This is not referenced in the text, and its purpose is not clear.

Section 3: Evolution of the international policy landscape

  • Section 3: This section reads as another list, with no clear contribution to the field being made, and no application to the question stated in the abstract
  • Line 410: As with the layers in section two, the phases presented in section three need justification and explanation
  • Lines 428-429: This knowledge should not be assumed. Detail how.
  • Line 432: Different formatting of ‘microplastic’ to previous uses
  • Table 2: This is poorly formatted (as well as being formatted differently to tables one and three) and not referenced in the text

Section 4: Where now? Strategic debates and new directions for global governance of plastics and plastic pollution

  • Section 4 is another list of examples that are not applied to a research question
  • Lines 509-515: The rest of the manuscript is not used to lead the reader to these statements, or to support them.
  • Table 3: The source of the information is not provided. There is no evidence to support what is included, indicating that this information is based on the opinions of the authors only.
  • Figure 2 is poorly formatted and not clear. The figure caption does not state what the figure shows, making a statement instead
  • Mention of the move to alternative materials misses the well-documented environmental impacts of alternative materials (See extensive literature on life-cycle assessments of plastics and alternative materials). Whilst the manuscript does not explicitly advise a move to alternatives, its framing of plastic alternatives as an alternative to plastics without consideration of these could be interpreted as such
  • Along these lines, I would like to have seen greater consideration of the argument that societies need to consume less overall. This manuscript reads as though the problem of plastic pollution is the material itself, with little consideration of the way society consumes.

Author Response

Response to Reviewer 1 .

Thank you for taking the time to read our paper so thoroughly and give us suggestions for improvements.  We have tried to incorporate them all, while also not making the paper too long and as you will see in the tracked version this involved a very significant re-writing of some parts of the paper and revisions of others.  The comments have really helped us clarify our presentation so thank you.   

  1. Re the main concern that it needs a clearer research question, identification of contribution to the field and transparent analytical framework – I hope we have addressed this now in the abstract, introduction, and at key points in the text (see the many track changes in revised version). However, if you feel it’s still not clear, please let us know, as we strongly want to do the work justice and to meet the expectations of readers. For one thing, we hope future researchers will follow up some of the on-going questions raised in the paper because the role of international policy with an upstream or life-cycle perspective is relatively new and has many research gaps. We hope it could become a more active research area in light of the new global trade treaty currently being negotiated, and the implications for implementation of the many low-carbon pledges recently made by governments. 
  2. Conclusions – I think we’ve now made a better job of leading up to the final argument that the plastic pollution issue needs to be addressed in a broader frame that incorporates a wider set of development, environmental, economic and trade concerns across the entire plastics life-cycle; that these need to be global rather than national, and cannot rely on voluntary measures.
  3. The Box, figures and tables have all been significantly re-worked and in one case deleted completely.  Formatting has not been easy for some reason and I hope the changes we made outside the template travel well in it. 
  4. The points that were not clear, or ill-defined or where you asked for further explanation have I hope now been improved.
  5. The challenge of finding alternatives to plastic – yes I agree, and we added a short clarification here but I did not make it too long as we wanted to keep attention onto the international policy theme
  6. Line 112, Yes you are correct, we can’t read our own handwriting, I think it is from page 5. I will check this with my co-author but for the moment am just changing the page number.
  7. Line 123. Yes, fair comment, we moved the Figure to a better place.
  8. Line 129 Yes I agree there are other extremely important causes that need to be considered, such as subsidization of virgin oil, linear business models etc. The subsidies issue comes up time and again and could really be an entire paper in itself.  At the same time, wowever we felt these fitted more into the discussions about production of plastics in the first place, not this section that focuses on waste management; so have cited them there.  There is so much one could say, and we feared making the paper too long, but thanks for drawing this to our attention.
  9. Line 164, re different opinions on the root causes of waste management problems in developing countries, we really just wanted to highlight the literature without going too far into it. We risk becoming terribly long in the paper and also deflecting attention from the role of international policy over plastics production, trade and consumption which is where we think we can add more value to the debate.  
  10. Line 195, Deleted this comment, agreed it is vague and also not essential for the paper.
  11. Line 228. We are trying to show that circular economy needs more than voluntary initiatives.  While these can be very helpful, it needs also mandatory ones and ideally these will be international policies as well as the national ones.  I have changed some of the headings and redrafted sections here to hopefully make that clearer. 
  12. Table 1. Fair comment, we tried not to be too exhaustive and just pick a selection for the table. Some of the ones you mention are however in the text and they are certainly important, so I have also added them into the table. Please suggest more explicitly if you think this will improve the message.
  13. Line 497, we wrote it as our “view” but you are right it is also a conclusion so I have re-written the later part of the paper to make it clearer what the conclusions are. Actually this reflects also the broader comments made in the beginning of your review, which I hope we’ve addressed.  If this is still not well enough stated please let us know because this message is important.
  14. Line 546. This is a really huge question and an area where we are currently extending the work. It is a big research gap for the world of plastics, although there is a big and relevant literature on industrial policy generally in the East Asian example, think of Alice Amsden for example (The Best and the Rest, among others). These studies focus on areas like coal and steel and ship-building, with some more recent literature on the move into light manufacturing and textiles, but not yet plastic.  It is too soon to give a proper answer to your question but I hope we can address that in a future article. Please do make suggestions or contact us about it later if you are interested in this topic.
  15. Line 591. Yes agreed this is an interesting and relevant point, do you have concrete examples or literature we should include? From my discussions with development banks I have not seen much interest or awareness about this, although they are aware of the need to invest in waste management systems which sometimes includes recycle facilities. Many development banks are now financing this, also Official Development Assistance. Is this another area that needs to be highlighted for future researchers?
  16. In the table of comments, you indicate that there is need to improve the references, do you mean throughout the text or that we have missed some important ones, or the formatting? Do please draw my attention to gaps, it is possible we have omitted important references or even that we do not yet know it, in which case it would be good to know so we can rectify.

Thank you again for your review, I hope we answered your questions and look forward to improving the paper.

Reviewer 2 Report

This report describes in great details the evolution of the global discourse on plastics and environmental burdens related to the over- and misuse of specific plastic products. It gives an excellent overview on on-going initiatives, their specific framing of plastics and potential solutions to the global plastic problem. Nevertheless it is not fit for publication in this journal because it is designed more as an opinion piece based and lacks a clear research question and a transparent analytical framework.

To a large extent the paper is rather descriptive and should aim to draw more conclusions e.g. on the observable or expected effectiveness of the different strategic approaches the paper describes. Currently it is difficult to say where the paper provides new scientific insights or how it contributes to the development of social sciences. In general the authors should be much more explicit what they see as conclusions from their analysis and what rather as subjective and often normative assumptions.

With regard to the overall framing of the problem the paper completely neglects the challenge of finding more sustainable alternatives to plastic. Phasing out plastics is compared to the transition towards a circular economy; neglecting that e.g. EU Commission´s Strategy on Plastics in a Circular Economy clear focuses on improved recyclability and closure of plastic flows, only in specific cases on phasing out plastics. E.g. in line 497 it says “it is the authors view …”; that very clearly describes the nature of the paper, these “views” should be backed up by scientific analysis.

Specific comments

Chapter 1 lacks a clear research question: What is the specific research gap this paper aims to address? To what extent does the paper contribute to the state of the art?

Line 112: Quotation seems to be wrong, page 3 of this report is just the preface.

Line 123: How does this figure on trade fit to the understanding or analysis of the three layers?

Line 129: Please be more transparent why just these two causes for plastic pollution have been selected and various other aspects like subsidisation of virgin oil, lacking recycling technologies, linear business models etc etc. are not taken into account.

Line 164: Please elaborate on this statement - what are the "root courses" and to what extent do they oppose the situation described above of lacking waste management infrastructures?

Line 195: How does this kind of very vague comment contribute to the paper?

Line 228: How does "circular economy" differ from above described voluntary initiatives? As described also the circular economy is based to a large extent on voluntary agreements.

Table 1: It´s unclear how this overview has been developed. Obviously important especially regulatory instruments are missing, e.g. mandatory recycled content quota, ecomodulated EPR fees, reuse quota...

Line 497: Is this an opinion or a conclusion?

Line 546: Definitely, but what can be learned and is it really transferable?

Line 591: What about investments into circularity of plastics...?

Author Response

Response to Reviewer 2.

Thank you for taking the time to read our paper so thoroughly and give us suggestions for improvements.  We have tried to incorporate them all, while also not making the paper too long.

  1. Re the main concern that it needs a clearer research question and transparent analytical framework – I hope we have addressed this now in the paper (see the many track changes in revised version). However, if you feel it’s still not clear, please let us know, as we strongly want to do the work justice and to meet the expectations of readers. For one thing, we hope future researchers will follow up some of the on-going questions raised in the paper because the role of international policy with an upstream or life-cycle perspective is relatively new and has many research gaps. This could become a very active research area in light of the new global trade treaty currently being negotiated, and the implications for implementation of the many low-carbon pledges recently made by governments. 
  2. Conclusions –I think we’ve now made a better job of leading up to the final argument that the plastic pollution issue needs to be addressed in a broader frame that incorporates a wider set of development, environmental, economic and trade concerns across the entire plastics life-cycle; that these need to be global rather than national, and cannot rely on voluntary measures.
  3. Re the challenge of finding alternatives to plastic, I agree this is a huge topic and incredibly important. We did feel that we’ve shown some of the challenges though, given the size and scale of the plastics industry as indicated through how massively it is traded and how this touches virtually all countries; also that is the reason we spend quite a lot of time in section 4 and 5. I don’t think that phasing out plastic is the same thing as transitioning to a circular economy; we see the circular economy efforts as complementing and helping, but not going so far as to phase out.  Its an ongoing debate,however; we don’t feel we have the final answers and rather want to show some light on what issues seem important in the international policy arena.  If we havn’t been clear on this please let us know and we will try to word it. 
  4. Research question – what is the specific research gap the paper aims to address, how does the paper contribute… I think this is well covered now in the new abstract and introduction, which are totally re-written. You were quite right to make this criticism and I hope we did a better job now.
  5. Line 112, Yes you are correct, we can’t read our own handwriting, re-checking it now online it is from page 5. I will check this with my co-author but for the moment am just changing the page number.
  6. Line 123. Yes, fair comment, we moved the Figure to a better place.
  7. Line 129 Yes I agree there are other extremely important causes that need to be considered, such as subsidization of virgin oil, linear business models etc. The subsidies issue comes up time and again and could really be an entire paper in itself.  At the same time, however we felt these fitted more into the discussions about production of plastics in the first place, not this section that focuses on waste management; so have cited them there.  There is so much one could say, and we feared making the paper too long, but thanks for drawing this to our attention.
  8. Line 164, re different opinions on the root causes of waste management problems in developing countries, we really just wanted to highlight the literature without going too far into it. We risk becoming terribly long in the paper and also deflecting attention from the role of international policy over plastics production, trade and consumption which is where we think we can add more value to the debate.  
  9. Line 195, Deleted this comment, agreed it is vague and also not essential for the paper.
  10. Line 228. We are trying to show that circular economy needs more than voluntary initiatives.  While these can be very helpful, it needs also mandatory ones if only because the industry is no enormous and has little incentive to change.  Ideally these will be international policies as well as the national ones, one reason being that companies can shift location and avoid them; also that impacts are cross-border tool.   I have changed some of the headings and redrafted sections here to hopefully make that clearer. 
  11. Table 1. Fair comment, we tried not to be too exhaustive and just pick a selection for the table. Some of the ones you mention are however in the text and they are certainly important, so I have also added them into the table. Please suggest more explicitly if you think this will improve the message.
  12. Line 497, we wrote it as our “view” but you are right it is also a conclusion so I have re-written the later part of the paper to make it clearer what the conclusions are. Actually this reflects also the broader comments made in the beginning of your review, which I hope we’ve addressed.  If this is still not well enough stated please let us know because this message is important.
  13. Line 546. This is a really huge question and an area where we are currently extending the work, as described in section 5. It is a big research gap for the world of plastics, although there is a big and relevant literature on industrial policy generally in the East Asian example, think of Alice Amsden for example (The Best and the Rest, among others). These studies focus on areas like coal and steel and ship-building, with some more recent literature on the move into light manufacturing and textiles, but not yet plastic.  It is too soon to give a proper answer to your question but I hope we can address that in a future article. Please do make suggestions or contact us about it later if you are interested in this topic because we hoping to pursue it soon.
  14. Line 591. Yes agreed this is an interesting and relevant point, do you have concrete examples or literature we should include? From my discussions with national and multilateral development banks I have not seen much interest or awareness about this, although a few seem aware of the need to invest in waste management systems which sometimes includes recycle facilities. Many development banks are now financing this, also Official Development Assistance. Is this another area that needs to be highlighted for future researchers?
  15. In the table of comments, you indicate that there is need to improve the references, do you mean throughout the text or that we have missed some important ones, or the formatting? Do please draw my attention to gaps, it is possible we have omitted important references or even that we do not yet know it, in which case it would be good to know so we can rectify.

Thank you again for your review, I hope we answered your questions and look forward to improving the paper.

Reviewer 3 Report

Title: Transforming the Global Plastics Economy: From concerns about pollution towards Development

Summary:

This article brings together an integrated analysis of the ‘political economy piece’ of evolving global discussions on challenges and responses to plastic pollution. First, the paper shows how the framing of the plastic pollution crisis has evolved. Second, the evolution of international policy with respect to plastics governance was highlighted. Third, future directions on global governance of plastics in international organizations such as the United Nations and World Trade Organisation was discussed. Finally, major gaps and future research was pointed out.

Comments:

The manuscript is written clearly with good logic with relevant background information about the topic. The aim of this study is to present past, current, and future status of plastic governance from a international perspective. The value of this article stands on the analysis of layers of plastics governance and the interrelations, making the gaps and challenges apparent.

The key issue of this article is the conceptual representation of different components of the analysis. Most of the figures and mind maps are lack of clearance due to small font or misplaced texts. Since these conceptual illustrations are key to this paper, I highly recommend revisions of them before publication.

Revisions:

Page

Line

Comment

Box1

-

Revision required.

Tables1-2

-

If possible, alignment to the left for easier reading

Figures

-

Please improve on the readability.

7

Foodnote 10

Different font

Author Response

Dear Reviewer 3, Thank you for the comments on our paper.  We have responded to them all.  Please see the attachment, and of course the changes in the revised text. thank you.

Reviewer 4 Report

The authors systematically deal with legal regulations regarding plastic waste and its negative impact on the environment, and if the amount of plastic waste is not reduced, what are the possible consequences. They also indicate what should be done in terms of reducing the use of plastics, especially packaging plastics.

This is an interesting issue, known to the public and a lot of research is being conducted about this problem, so I think there is no significant news. However, the authors could give the short overview of what has been achieved, for example by reducing the use of plastic bags and the like and what type of plastic packaging could be completely excluded or replaced by an alternative (the authors did mention but not specifically).

In my opinion, clear guidelines should be given as to which type of plastic use (packaging, bags) will be reduced and what benefits will be achieved for the environment because it is clear that plastic packaging will not be able to be completely replaced or banned in the near future.

I ask myself and you why the use of glass packaging would not increase in the future because it is more environmentally friendly and easier to recycle compared to plastic.

On the other hand, from year to year the amount of plastic waste is growing and at the same time the legal regulations are becoming stricter with bans on certain types of plastic packaging or bags ...

Specific comments:

All abbreviations used in the text must be explained with the the full name when first mentioned.

Below Box 1 it seems unnecessary to me to write: "Source: Authors"

Line 180: „micro-organisms that can eat plastics“ – „eat“ or decompose

Figure caption should be below the Fig.

I think there is no need to use "covid" in keywords - there is no basis, it is mentioned 2 times in the text, but not in the context of the issue of covid disease or any significance to plastic waste.

Author Response

Dear Reviewer 4, Thank you for taking the time to read and help improve our paper.  We responded to all your points as you will see in the revised version of the text.  Also please see the attachment. Kind regards and thank you.

Round 2

Reviewer 1 Report

Whilst extensive edits have been made to this manuscript, it still fails to meet the requirements of a research article. Its reference in the abstract to literature reviews and empirical findings necessitate a methods section that is lacking and, though the empirical database it refers to is cited it is not used well throughout the manuscript to support the points that are made.

Across the abstract and introduction, six 'aims' are introduced. Some of these are similar, others are not. It is still not clear what knowledge gap the authors are trying to fill, or what the focus of this very busy manuscript is.

There are several assumptions about the environmental problems associated with plastic pollution, particularly in comparison to plastic alternatives. The problem of plastic pollution remains undefined by the authors, and the authors do not consider the relative environmental impacts of some of the changes to behaviour and practice that they suggest.

There is no research, novel or otherwise, presented in this manuscript, and therefore it must not be published as a research article. There is no way that another author could attempt to reproduce the article because there is no experimentation.

Though there is a clear understanding of the legal and policy context of this manuscript, the environmental context demonstrates a naivety that undermines the authority of the former. I disagree with the comments from the other reviewers that the manuscript is written with 'relevant background'. The authors do not demonstrate that they have any understanding of the environmental impacts of plastic pollution, or how they are situated within broader discourses of the environmental harm of consumption and pollution. It reads as though they have heard that plastic pollution is a problem, but have not read into it themselves. Discussing economic and political policy without this context is dangerous.

This manuscript is purely based on opinion from a synthesis of literature and some data, the integrity of which cannot be verified from the manuscript. It is an opinion / perspective piece, and should be written up as such if it is to be published at all.

Author Response

Dear Reviewer 1, Thank you for taking the time to go through our paper again.  We are taken on board your comments, especially the concern about the treatment of pollution because we did not intend the paper to give that impression.  Similarly I noted your concern that subsequent researchers would need to know how to trace our path.  We have revised the text accordingly. The overview of what we did is in the attachment.  I am not sure how we can send the revised paper however, it does not seem to be possible in this template.   thank you once again, regards, the authors.
